# Shaping and interpretation of Dpp morphogen gradient by endocytic trafficking

**Sheida Hadji Rasouliha, Gustavo Aguilar, Cindy Reinger, Shinya Matsuda**¤*

Biozentrum, University of Basel, Basel, Switzerland

¤ Current address: Department of Biological Sciences, Graduate School of Science, The University of Tokyo, Tokyo, Japan
* shinyamatsuda0423@gmail.com

## Abstract

Dpp/BMP is a morphogen that controls the patterning and growth of the Drosophila wing disc. Although endocytic trafficking has been proposed to influence both extracellular Dpp distribution and signaling, how this process shapes and interprets the Dpp gradient under physiological conditions remains unclear due to limitations in visualizing endogenous Dpp. Here, we generated fluorescent protein-tagged functional *dpp* alleles that allow simultaneous visualization of extracellular and intracellular Dpp distributions. Using these tools, we found that, while Dynamin-mediated internalization is required for Dpp signaling activation, Rab5-mediated early endosomal trafficking is dispensable for Dpp spreading and signaling initiation but is required for signal termination by promoting the downregulation of activated receptors. We show that Dpp signaling is terminated at the multivesicular body (MVB), likely through ESCRT-dependent sorting of activated receptors into intraluminal vesicles (ILVs), rather than via Rab7-mediated lysosomal degradation. Notably, blocking MVB formation expanded the Dpp signaling gradient without altering the extracellular Dpp gradient, thus compromising extracellular Dpp gradient interpretation. Together, our findings reveal that the extracellular Dpp gradient is shaped by Dynamin-dependent internalization and interpreted through the duration of intracellular signaling.

## Author summary

During development, cells need instructions to form organs and tissues with the right size and shape. These instructions often come from signaling molecules called morphogens, which travel through tissues in gradients. One such morphogen, called Dpp, helps shape the fruit fly wing by telling cells where they are and what to become. To better understand how cells respond to Dpp, we created fruit flies that produce a fluorescent version of the molecule, allowing us to track this molecule both outside and inside cells. We found that cells must uptake

**Data availability statement:** All relevant data are within the manuscript and its Supporting Information files.

**Funding:** G.A. was supported by "Fellowships for Excellence" from the International PhD Program in Molecular Life Sciences of the Biozentrum, University of Basel. S.M. was supported by a SNSF Ambizione grant (PZ00P3_180019). The funders had no role in study design, data collection and analysis, decision to publish, or preparation of the manuscript.

**Competing interests:** The authors have declared that no competing interests exist.

Dpp from their surface to start the signaling. The signal is terminated when Dpp reaches a specific compartment inside the cell. When this termination step is blocked, cells continue responding to Dpp longer than they should—even when the amount of Dpp outside the cells remains unchanged. This shows that cells respond not only to how much Dpp they see on the cell surface, but also to how long they are exposed to it inside cells. Our findings help explain how cells use these signals to determine where they are and what they should become during development.

## Introduction

Morphogens are signaling molecules produced by a localized group of cells that regulate the fate of neighboring cells in a concentration-dependent manner [1]. Among these, Decapentaplegic (Dpp)—the Drosophila homologue of vertebrate bone morphogenetic proteins 2 and 4 (BMP2/4)—has served as a valuable model for studying morphogen function. Dpp is produced by a stripe of cells in the anterior compartment along the anterior-posterior (A/P) compartment boundary of the Drosophila wing imaginal disc. From this source, Dpp spreads and forms a concentration gradient to regulate both tissue patterning and growth [2–6].

Based on the severe patterning and growth defects observed in *dpp* mutant flies [7–9], it has long been thought that Dpp spreading from the stripe is essential for these functions. However, we recently showed that blocking Dpp spreading from the source cells had only a minor impact on the anterior compartment, while severely disrupting posterior patterning and growth. These results suggest that Dpp spreading is not critical for the overall patterning and growth but is primarily required for posterior patterning and growth [10,11].

Although Dpp spreading-mediated gradient formation is less critical than previously thought, the extracellular Dpp gradient is still established by a variety of extracellular and cell surface molecules and interpreted in the nucleus [4–6]. Dpp is thought to bind to the Type I and Type II receptors, Tkv and Punt, on the cell surface, which induces the phosphorylation of Mad (pMad) in the target cells. pMad is then translocated into the nucleus, where it regulates the expression of Dpp target genes, primarily by repressing Brk, a repressor of Dpp target genes [4]. Thus, the extracellular Dpp gradient is converted into a nuclear pMad gradient, which inversely shapes the Brk gradient. These two opposing gradients regulate the nested expression of target genes, specifying the positions of future adult wing veins, such as L2 and L5, as well as promoting growth [12–14].

In addition to extracellular regulation and nuclear interpretation, endocytic trafficking has also been implicated in shaping and interpreting gradients of different morphogens [15–18]. However, how the extracellular Dpp morphogen gradient is shaped and interpreted through endocytic trafficking remains unclear. Several models have been proposed regarding the role of endocytosis in shaping the Dpp morphogen gradient. First, since Dpp accumulates in *tkv* mutant clones, particularly in cells close to

the source cells, it was hypothesized that Dpp is internalized and transported by Tkv through repeated cycles of endocytosis and exocytosis [19]. Second, it has recently been proposed that heparan sulfate proteoglycans, such as Dally, rather than Tkv, act as cell-surface receptors to internalize and recycle Dpp, contributing to the extracellular Dpp gradient [20]. In this model, Dpp is thought to bind to Tkv intracellularly to activate Dpp signaling. While both models have been challenged [21–24], endocytic trafficking may influence Dpp spreading through other cell surface factors. Third, Tkv-mediated endocytosis has been proposed to act as a sink to remove extracellular Dpp [23–25], while Dally antagonizes this process to establish a long-range Dpp gradient [24,25].

Thus, while the role of extracellular and cell-surface factors in regulating the extracellular Dpp gradient is well established, the impact of endocytic trafficking on the gradient itself remains controversial. Additionally, the mechanisms by which the extracellular Dpp gradient is interpreted at the cellular level are still unclear. Interestingly, it has been shown that Dpp predominantly exists intracellularly, and Dpp signaling is lost in endocytosis-deficient cells [26–29], highlighting the importance of internalized Dpp for signal activation. However, it remains unclear in which endocytic compartment Dpp signaling is activated or terminated, and whether the duration of Dpp signaling influences the interpretation of the gradient. Recently, fluorophore-conjugated anti-GFP nanobodies were used to label and trace only internalized GFP-Dpp [20], but it remains unclear whether the nanobody-bound GFP-Dpp is functional. Therefore, the role of endocytosis in Dpp gradient formation and interpretation remains unclear partly due to the lack of suitable *dpp* alleles that would allow visualization of both extracellular and intracellular Dpp distribution at the physiological level.

In this study, we generated functional fluorescent protein-tagged *dpp* alleles and systematically investigated the role of endocytic trafficking in Dpp morphogen gradient formation and Dpp signaling activity. We first confirmed that blocking Dynamin-mediated endocytosis expanded the extracellular Dpp distribution but impaired Dpp signaling [23]. Surprisingly, in contrast to earlier reports [19,30], we then found that blocking Rab5-mediated early endosome formation expanded the range of Dpp signaling likely due to impaired downregulation of Tkv. These results indicate that while Dpp signaling initiation depends on Dynamin-mediated internalization, its attenuation is regulated through Rab5-dependent endocytic trafficking. Furthermore, we showed that blocking multivesicular body (MVB) formation, but not Rab7-mediated lysosomal degradation, expanded intracellular Dpp distribution and Dpp signaling range, while leaving the extracellular Dpp gradient unaffected. These results indicate that Dpp signaling is terminated at the MVB, which is critical for translating the extracellular Dpp gradient into a Dpp signaling gradient. Taken together, these results suggest that the extracellular Dpp gradient is shaped by Dynamin-mediated internalization and interpreted through the duration of intracellular Dpp signaling.

## Results

### Visualization of extracellular and intracellular Dpp gradient in the wing disc

We previously generated an endogenous GFP-dpp allele by inserting GFP after the last processing sites of Dpp to tag the mature Dpp [10]. However, the resulting GFP-Dpp fluorescent signal was too weak to visualize the graded distribution (Fig 1A), consistent with findings from another independently generated GFP-dpp allele [20]. To better visualize the endogenous Dpp gradient, we then inserted either mGreenLantern (mGL) [31] or mScarlet (mSC) [32] into the *dpp* locus, generating *mGL-dpp* and *mSC-dpp* alleles, respectively. Notably, mGL-Dpp produced a much brighter fluorescent signal than GFP-Dpp (Fig 1A and 1B) and revealed a clear graded distribution outside the stripe of Dpp source cells (Fig 1B). A similar graded distribution was also observed with mSC-dpp fluorescent signal (Fig 1C).

Unlike the *GFP-dpp* allele, the two newly generated alleles were not haploinsufficient but were semi-lethal. To overcome the partial embryonic lethality, we introduced a transgene known as "JAX", which contains the genomic region of *dpp* essential for early embryogenesis [33] but does not rescue the wing phenotypes of *dpp* mutants [24]. We found that the lethality associated with each allele was substantially rescued by JAX, resulting in homozygous viable flies without obvious phenotypic defects (Fig 1D–1F). JAX did not affect Dpp signaling in functional *HA-dpp* wing discs [10] (Fig 1G,

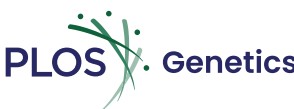

**Fig 1. Visualization of the endogenous Dpp morphogen gradient in the wing disc.** (A) GFP-Dpp fluorescent signal from *GFP-dpp/Cyo, p23* wing disc. (A), mGL-Dpp fluorescent signal from *mGL-dpp/* +wing disc (B), and mSC-Dpp fluorescent signal from *mSC-dpp/* +wing disc (C). (D) Adult wing of yw (Control). (E) Adult wing of *JAX; mGL-dpp/mGL-dpp*. (F) Adult wing of JAX; *mSC-dpp/mSC-dpp*. (G-J) α-pMad staining of *HA-dpp/HA-dpp* (G), *JAX; HA-dpp/HA-dpp* (H), *JAX; mGL-dpp/mGL-dpp* (I), and *JAX; mSC-dpp/mSC-dpp* wing disc (J). (K) Average fluorescence intensity profile of (G-J). Data are presented as mean +/- SD. (L) mGL-Dpp fluorescent signal (L), extracellular α-GFP staining (L'), and merge (L") of *mGL-dpp/* +wing disc. Scale bar: 50μm.

1H and 1K), and Dpp signaling levels were comparable among *JAX;HA-dpp*, *JAX;mGL-dpp* and *JAX;mSC-dpp* wing discs (Fig 1H–1K). These results indicate that both *mGL-dpp* and *mSC-dpp* alleles are functional, at least during wing disc development.

To determine whether the mGL-Dpp fluorescent signal originates from extracellular or intracellular Dpp, we performed extracellular staining using an α-GFP antibody, which recognizes mGL (like GFP, as both are derived from *Aequorea victoria*) and does not interfere with GFP fluorescence [34]. By comparing the extracellular mGL-Dpp distribution with the total mGL-Dpp fluorescence, we found that while the total mGL-Dpp signal was highest at the center of the wing disc, corresponding to the site of *dpp* expression (Fig 1L), the extracellular mGL-Dpp displayed a shallower graded distribution (Fig 1L'). The elevated total signal in the center likely reflects intracellular Dpp within the secretory pathway of Dpp-producing cells. Interestingly, the Dpp signaling gradient appears steeper than the extracellular mGL-Dpp gradient, suggesting a threshold-dependent response to extracellular Dpp required for signaling activation.

Interestingly, the extracellular α-GFP distribution and total mGL-Dpp fluorescence rarely colocalized (Fig 1L"). Since the total mGL-Dpp fluorescent signal was not sensitive to acid wash, which efficiently removes extracellular proteins [20] (S1 Fig), the majority of the total mGL-Dpp fluorescent signal likely originates from intracellular Dpp. Furthermore, the extracellular staining amplifies weak or even sub-detection levels of the total mGL fluorescence to reveal extracellular Dpp distribution. Together, these factors may explain the limited overlap between the mGL fluorescence and the extracellular α-GFP signal.

To determine the subcellular localization of endogenous Dpp, we compared mSC-Dpp distribution with various Rab proteins tagged with eYFP (Fig 2). Mander's coefficient (M1) analysis revealed that mSC-Dpp showed varying degrees of colocalization with the early endosome marker Rab5-eYFP (Fig 2A'), the late endosome marker Rab7-eYFP (Fig 2B'), the fast-recycling endosome marker Rab4-eYFP (Fig 2C'), and the slow-recycling endosome marker Rab11-eYFP (Fig 2D'). These results indicate that internalized Dpp is trafficked through multiple endocytic compartments.

## Rab5 is required for downregulating Dpp signal

To study how different endocytic compartments contribute to Dpp gradient formation and signaling, we first knocked down Dynamin GTPase (*Drosophila* homologue: shibire), a critical factor required for vesicle scission from the plasma membrane [35]. Consistent with the idea that Dpp signaling is activated upon endocytosis [23], we found that the temperature-sensitive shibire allele (*shi^ts1^*) resulted in a complete loss of Dpp signaling after 2 hours at the restrictive temperature (Fig 3A–3C).

Previous studies have shown that loss of Rab5, either through a dominant-negative form or RNAi, reduces Dpp signaling and the expression of its target genes, suggesting that Dpp is transported via endocytosis [19] and/or that Dpp signaling is activated at or downstream of the early endosome [30]. To test this, we used *ap-Gal4* combined with the temperature-sensitive Gal80 (Gal80ts), hereafter referred to as *ap^ts^* for simplicity, to temporally express UAS-Rab5 RNAi in the dorsal compartment of the wing discs (*ap^ts^>Rab5RNAi*) (Fig 3D). To minimize the pleiotropic effects associated with Rab5 knockdown, Gal80ts enables precise temporal control of gene expression through temperature shifts: at 18°C, Gal80ts binds and inhibits GAL4 activity, while at 29°C, Gal80ts becomes inactive, allowing Gal4 to drive transcription from UAS sites. Under this system, the ventral compartment serves as an internal control.

In stark contrast to previous findings, we observed that temporal knockdown of Rab5 by RNAi (*ap^ts^>Rab5RNAi*) led to a marked increase in Dpp signaling activity relative to the control ventral compartment (Fig 3D–3F). Similar results were obtained with independent Rab5 RNAi lines as well as a dominant-negative form of Rab5 line (Fig 3G–3J). Furthermore, induction of *rab5* null clones (*rab5²*) [36] resulted in a cell-autonomous reduction of Brk expression (Fig 3K–K'), consistent with enhanced Dpp signaling, as Brk is repressed by Dpp signaling. Together, these findings suggest that Rab5 is not required for the activation of Dpp signaling but instead plays a key role in its downregulation.

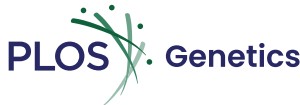

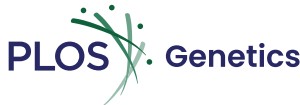

**Fig 2. Colocalization of mSC-Dpp with different Rabs.** (A-D) Comparison of mSC-Dpp with Rab5-eYFP (A), Rab7-eYFP (B), Rab4-eYFP (C), Rab11-eYFP (D) in the late third instar wing discs. Mander's coefficient (M1) indicates the percentage of overlap of mSC-Dpp with different Rabs. Scale bar: 20μm.

## The effects of Rab5 on Dpp distribution

To investigate how loss of Rab5 leads to increased Dpp signaling, we first asked whether this increase is dependent on Dpp itself. Because *dpp* mutants lack wing tissue, we took advantage of the fact that Dpp promotes wing disc growth primarily by repressing the growth inhibitor Brk, and that *dpp, brk* double mutants can grow in the absence of Dpp signal [37–39]. When Rab5 was knocked down in a *dpp, brk* mutant background, Dpp signaling was not upregulated, indicating that the observed increase in Dpp signaling was indeed Dpp-dependent (Fig 4A). Next, to test whether this effect was due to changes in *dpp* transcription, we monitored a *dpp-lacZ* transcriptional reporter following Rab5 knockdown and found no change in reporter activity (Fig 4B), indicating that transcriptional upregulation of *dpp* is not responsible.

**Fig 3. Rab5 is required for downregulating Dpp signaling.** (A-B) α-pMad staining of *shi^ts^/+* wing disc (control) (A) and *shi^ts^* wing disc (B) upon 2h at restrictive temperatures. (C) Average fluorescence intensity profile of (A, B). Data are presented as mean+/- SD. (D, E) α-pMad staining (D) and α-Rab5 staining (E) of *ap^ts^ > rab5RNAi* (30518) wing disc. (F) Average fluorescence intensity profile of (D). Data are presented as mean+/- SD. (G-J) α-pMad staining of *ap^ts^ > rab5RNAi* (34096) wing disc (G), *ap^ts^ > rab5RNAi* (103945) wing disc (H), *ap^ts^ > rab5DN* (42703) wing disc (I), and *ap^ts^ > rab5DN* (42704) wing disc (J). (K) *rab5²* null clones generated in the peripheral regions of the wing disc visualized via absence of α-β-gal staining (K) and α-Brk staining (K'). Scale bar: 30µm.

We then asked whether changes in Dpp distribution contribute to altered Dpp signaling. Consistent with impaired endocytosis in *rab5* mutants [40], loss of Rab5 led to an accumulation of extracellular mGL-Dpp, particularly outside of the Dpp producing cells (Fig 4C–4E), similar to what was observed in *shi^ts1^* mutants (Fig 4F–4H). Cross-sectional imaging of wing imaginal discs revealed increased extracellular mGL-Dpp along the basolateral side (Fig 4D, yellow arrowheads).

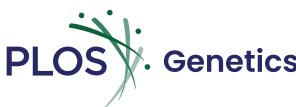

**Fig 4. Changes in Dpp distribution in the absence of Rab5.** (A) α-Rab5 staining (A) and α-pMad staining (A') of *brk, dpp^{d8}/dpp^{d12}, ap^{ts} > Rab5 RNAi* (30518) wing disc. (B) α-Rab5 staining (B), α-pMad staining (B'), and α-β-gal staining (B") of *dpp-lacZ/ +, ap^{ts} > Rab5 RNAi* (30518) wing disc. (C) Extra-cellular α-GFP staining of *mGL-dpp/ +, ap^{ts} > Rab5 RNAi* (30518) wing disc. (D) Optical cross-section of (C). (E) Average fluorescence intensity profile of

(C). Data are presented as mean +/- SD. (F, G) Extracellular α-GFP staining of *shi^ts/+, mGL-dpp/+* wing disc (F) and *shi^ts, mGL-dpp/+* wing disc (G) after 2h at restrictive temperature of 34°C. (H) Average fluorescence intensity profile of (F, G). Data are presented as mean +/- SD. (I, J) mGL-Dpp intracellular fluorescent signal of the lateral side (I) and basal side (J) of *mGL-dpp/+, ap^ts>rab5 RNAi* (30518) wing disc. (K, L) Tkv-YFP fluorescent signal of lateral side (K) and basal side (L) of *tkv-YFP/+, ap^ts>rab5 RNAi* (30518) wing disc. (M, N) Comparison of the number of puncta of (M, N). Two-sided Mann−Whitney test was used for the comparison; p = 0.1143 (n = 4) (M), p = 0.0067 (n = 8) (N). (O, P) Average fluorescence intensity profile of (K, L). Data are presented as mean +/- SD. Scale bar: 30μm.

While this increase of extracellular Dpp could potentially enhance Dpp signaling, Dpp signaling was lost in *shi^ts1* mutants despite similar extracellular Dpp accumulation (Figs 3B and 4F–4H). This indicates that Dynamin-mediated endocytosis is required for the activation of Dpp signaling. Therefore, the increase in extracellular Dpp upon Rab5 loss is unlikely to be the direct cause of the observed increase of Dpp signaling.

Given that internalized Dpp is required for signaling activation (Figs 3B and 4F–4H), we next asked whether intracellular Dpp distribution is altered upon loss of Rab5. As expected from impaired endocytosis, Rab5 knockdown appeared to reduce the number of intracellular mGL-Dpp puncta laterally, although this reduction was not statistically significant (Fig 4I and 4M). In contrast, Rab5 loss led to a significant increase in mGL-Dpp puncta at the basal side of the wing disc (Fig 4J and 4N). Since endocytic trafficking downstream of Rab5 is also disrupted in Rab5 mutants [36,40,41], mGL-Dpp likely accumulates in early endocytic vesicles that fail to fuse with early endosomes. Similarly, Tkv-YFP showed reduced lateral distribution particularly in the posterior compartment and accumulated basally in peripheral regions upon Rab5 knockdown (Fig 4K, 4L, 4O and 4P).

## Increase of Dpp signaling by loss of Rab5 is dependent on excess Tkv

We hypothesized that the intracellular accumulation of Dpp and Tkv upon Rab5 loss may contribute to the observed increase in Dpp signaling. If so, reducing excess Tkv should suppress the elevated pMad levels caused by Rab5 knockdown. To test this, we employed deGradHA, a genetically encoded system for targeted degradation of HA-tagged proteins [42]. Since Tkv is the key receptor for Dpp signaling, we used deGradHA to selectively degrade one copy of Tkv-HA-eGFP in the dorsal compartment of the wing disc (Fig 5). Under these conditions, this manipulation had minimal effect on the Dpp signaling gradient, aside from a slight reduction along the anterior-posterior (A/P) compartment boundary—likely due to the naturally low levels of Tkv in that region (Fig 5D)—indicating that deGradHA effectively removes only excess Tkv without broadly impairing signaling.

As expected, Dpp signaling was comparable between the dorsal and ventral compartments in control discs (Fig 5A). Knockdown of Rab5 via RNAi in the dorsal compartment resulted in elevated Dpp signaling relative to the ventral compartment (Fig 5B). However, simultaneous loss of Rab5 and degradation of one copy of Tkv-HA-eGFP in the dorsal compartment restored Dpp signaling to levels comparable to the ventral side (Fig 5C). These results support the idea that the increased Dpp signaling caused by Rab5 loss is dependent on excess Tkv, likely resulting from impaired endocytic trafficking.

## Loss of ESCRT components increases Dpp signaling without affecting extracellular Dpp gradient

Our results so far suggest that Dpp signaling is terminated through Rab5-mediated trafficking of Dpp and Tkv. But in which endocytic trafficking compartment is Dpp signal terminated downstream of Rab5? As early endosomes mature into late endosomes, the ESCRT (Endosomal Sorting Complex Required for Transport) machinery recognizes and sorts ubiquitinated proteins into intraluminal vesicles (ILVs), leading to the formation of multivesicular bodies (MVBs). These MVB-containing late endosomes subsequently fuse with lysosomes to degrade their contents [43].

Previous studies have proposed that Dpp signaling is terminated via endosomal degradation of activated Tkv [44,45]. Supporting this model, RNAi-mediated knockdown of factors required for MVB formation—such as the ESCRT-II



**Fig 5. Partial degradation of Tkv rescues increase of Dpp signaling upon loss of Rab5.** (A-D) Tkv-HA-eGFP fluorescent signal (A-D) and α-pMad staining (A'-D') of *tkv-HA-eGFP/+, ap^{ts}>+* wing disc (Control) **(A)**, *tkv-HA-eGFP/+, ap^{ts}>rab5 RNAi* (30518) wing disc (B), *tkv-HA-eGFP/+, ap^{ts}>rab5 RNAi* (30518)*, deGradHA* wing disc (C) and *tkv-HA-eGFP/+, ap^{ts}>deGradHA* wing disc (D). (A"-D") Average fluorescence intensity profiles of (A'-D'). Data are presented as mean +/- SD. Scale bar: 50μm.

component TSG101, the ESCRT-III component Shrub, or Vps4—in the dorsal compartment led to an increase in both the intensity and range of Dpp signaling compared to the ventral side (Fig 6A–6C). Consistent with the impaired sorting of ubiquitinated receptors into ILVs, knockdown of Vps4, Shrub, or Tsg101 resulted in the accumulation of Tkv and ubiquitin as large puncta that strongly colocalized (Figs 6D and S2). In contrast, such colocalization was not observed upon loss of Rab7 (Fig 6E), likely because Rab7 is not required for MVB biogenesis [46].

We then tested whether blocking MVB formation affects Dpp distribution. Similar to Tkv accumulation, intracellular mGL-Dpp also accumulated as large puncta upon loss of Vps4 (Fig 6F and 6G). Consistent with the accumulation of

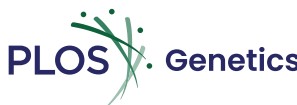

**Fig 6. Loss of ESCRT components increases Dpp signaling without affecting extracellular Dpp gradient.** (A-C) α-pMad staining of *ap^ts^>tsg101 RNAi* wing disc (A), *ap^ts^>shrub RNAi* wing disc (B), and *ap^ts^>Vps4 RNAi* wing disc (C). (D) Tkv-YFP fluorescent signal (D), α-Ubiquitin staining (D'), and colocalization map (D") of *tkv-YFP/+, ap^ts^>vps4 RNAi* wing disc. While 80.3% of TkvYFP co-localized with α-Ubiquitin in the dorsal compartment, 56.2% of TkvYFP co-localized with α-Ubiquitin in the ventral compartment. (E) Tkv-YFP fluorescent signal (E), α-Ubiquitin staining (E'), and colocalization map (E") of *tkv-YFP/+, ap^ts^>rab7 RNAi* wing disc. While 1.4% of TkvYFP co-localized with α-Ubiquitin in the dorsal compartment, 0.9% of TkvYFP co-localized with α-Ubiquitin in the ventral compartment. (F) mGL-Dpp fluorescent signal from the apical side (F), with magnified region of the dorsal compartment (F'), and the ventral compartment (F") of *mGL-dpp/+, ap^ts^>Vps4 RNAi* wing disc. (G) Comparison of the size of mGL-Dpp puncta between

F' and F''. Two-sided Mann−Whitney test was used for the comparison ($p = 0.0317$) (n = 5). (H) Rab5-eYFP fluorescent signal (H), mSC-Dpp fluorescent signal (H'), and merge (H'') at the basal side of *mSC-dpp/ +, ap^{ts}>tsgs101 RNAi* wing disc. (I) Comparison of % of mSC-Dpp puncta overlapping with Rab5-eYFP between dorsal and ventral compartment (H). Two-sided Mann−Whitney test was used for the comparison ($p = 0.0002$) (n = 13). (J) Extracellular α-GFP staining of *mGL-dpp/ +, ap^{ts} > Vps4 RNAi* wing disc. (K) Average fluorescence intensity profiles of (J). Data are presented as mean +/- SD.

ubiquitinated proteins in Rab5-positive compartments [44], we observed increased co-localization of mSC-Dpp with Rab5-eYFP following TSG101 knockdown by RNAi (Fig 6H and 6I). These results suggest that Dpp and Tkv accumulate in early endosomes and contribute to enhanced Dpp signaling when MVB formation is blocked. Interestingly, despite the expanded Dpp signaling gradient and accumulation of intracellular Dpp (Fig 6A–6C and 6F–6I), the extracellular mGL-Dpp gradient remained unchanged upon Vps4 knockdown (Fig 6J and 6K). This indicates that the extracellular Dpp gradient is translated into the intracellular Dpp signaling gradient through MVB-mediated downregulation of signaling components.

## Late endosomal trafficking is dispensable for terminating Dpp signaling

The increase in Dpp signaling upon the loss of MVB formation suggests that Dpp signaling is terminated either at the level of MVBs or through Rab7-mediated lysosomal degradation. Since Rab7 loss did not affect the distribution of ubiquitinated protein (Fig 6E), MVB formation is likely intact in *rab7* mutants. This allows us to distinguish the roles of MVB formation versus Rab7-dependent lysosomal degradation in regulating Dpp distribution and signaling. Surprisingly, immunostaining revealed that both clones of cells null mutant for *rab7* [47] and temporal knockdown of Rab7 by RNAi in the dorsal compartment significantly reduced Rab7 protein levels but had no effect on Dpp signaling activity (Fig 7A–7D). Furthermore, neither the extracellular nor intracellular mGL-Dpp distribution was altered upon Rab7 knockdown (Fig 7E–7H). These results suggest that MVB formation, rather than Rab7-mediated lysosomal degradation, is essential for terminating Dpp signaling.

## Recycling endosome is largely dispensable for extracellular Dpp gradient formation and signaling

While loss of Dynamin or Rab5 expanded the extracellular Dpp gradient (Fig 4), loss of ESCRT components or Rab7 had no effect on it (Figs 6 and 7). These results raise the question of whether endocytic trafficking beyond the early endosome contributes to the formation of the extracellular Dpp gradient. A recent study used a fluorophore-conjugated anti-GFP nanobody to selectively trace internalized GFP-Dpp [20]. Using this approach, internalized GFP-Dpp was shown to recycle back to the extracellular space, and knockdown of Rab4 or Rab11—key regulators of recycling endosomes—markedly reduced the distribution of overexpressed GFP-Dpp [20]. These results suggest that recycling endosomes play a crucial role in shaping the extracellular Dpp gradient.

To test this under more physiological conditions, we knocked down Rab4 or Rab11 using the same RNAi lines and examined both Dpp signaling and extracellular mGL-dpp distribution. Contrary to the previous study [20], knockdown of Rab4 by RNAi had no significant effect on Dpp signaling or extracellular mGL-Dpp distribution, aside from a slight decrease on the basal side (Fig 8A–8E). Similarly, Rab11 knockdown did not alter Dpp signaling or extracellular mGL-Dpp distribution (Fig 8F–8J). To test whether Rab4 and Rab11 function redundantly, we simultaneously knocked down both genes. However, even in this condition, neither Dpp signaling nor extracellular mGL-Dpp distribution was significantly affected (S3 Fig).

We also examined the effects of Rab4 or Rab11 loss on the intracellular mGL-Dpp distribution. Interestingly, while the overall number of intracellular mGL-Dpp puncta remained unchanged upon loss of Rab4 (Fig 8K and 8L), knockdown of Rab11 led to the accumulation of large intracellular mGL-Dpp puncta, particularly on the basal side (Fig 8M and 8N). This may reflect the formation of enlarged early endosomes, a phenotype previously reported upon loss of Rab11 [48]. Taken together, these results suggest that Rab4- and Rab11-mediated recycling endosome formation is not essential for establishing the extracellular Dpp gradient or Dpp signaling gradient.

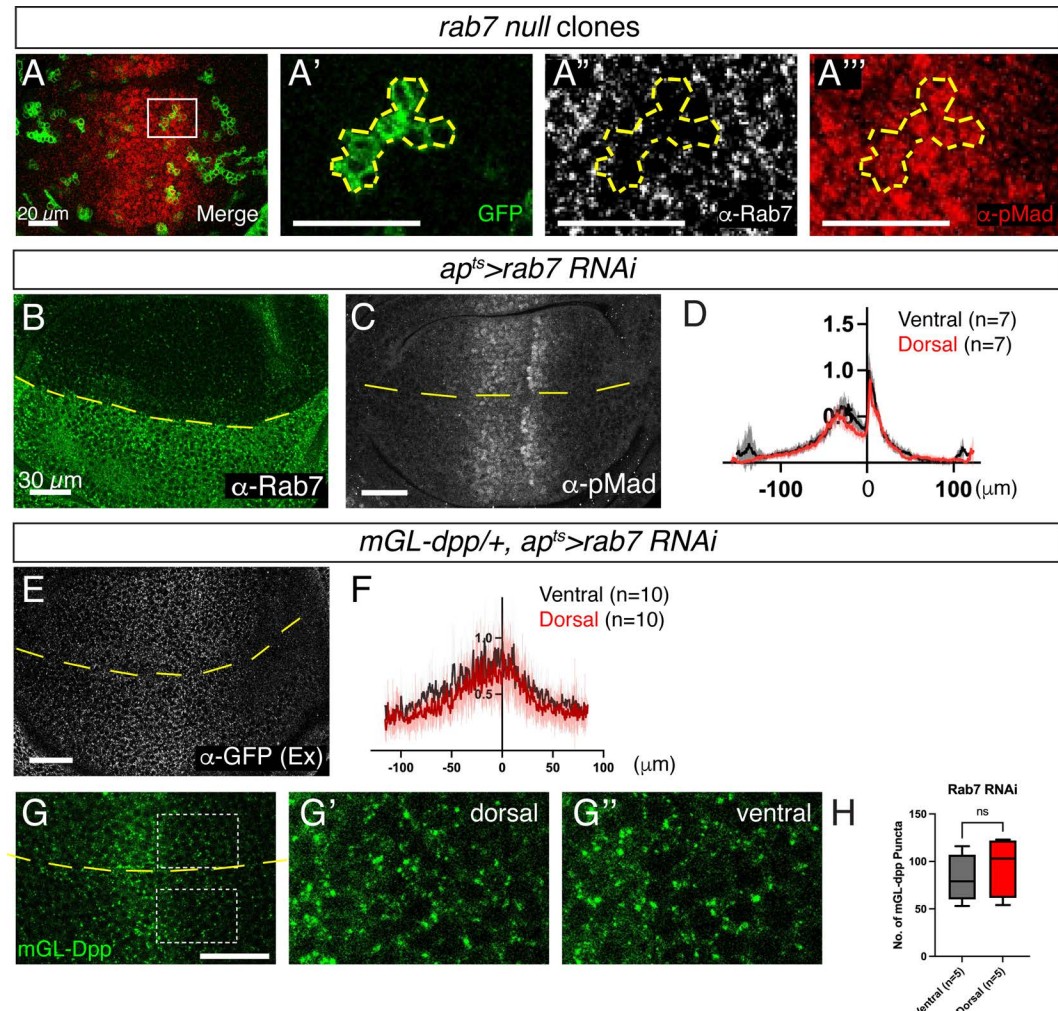

**Fig 7. Late endosomal trafficking is not involved in terminating Dpp signaling.** (A) Merge (A), GFP signal (A'), α-Rab7 staining (A"), and α-pMad staining (A"') of rab7 null clones (labeled by GFP signal) generated by MARCM. (B, C) α-Rab7 staining (B) and α-pMad staining (C) of *ap*ᵗˢ*>rab7 RNAi* wing disc. (D) Average fluorescence intensity profiles of (C). Data are presented as mean+/- SD. (E) Extracellular α-GFP staining of *mGL-dppl/ +, apᵗˢ>rab7 RNAi* wing disc. (F) Average fluorescence intensity profiles of (E). Data are presented as mean+/-SD. (G) mGL-Dpp fluorescent signal from apical side (G), magnified region in the dorsal compartment (G'), and magnified region in the ventral compartment (G") of *mGL-dppl/ +, apᵗˢ>rab7 RNAi* wing disc. (H) Comparison of the number of mGL-Dpp puncta between G' and G". Two-sided Mann−Whitney test was used for the comparison ($p = 0.4206$).

## Discussion

In this study, we generated novel *dpp* alleles that allow visualization of both extracellular and intracellular Dpp distributions under the physiological conditions. Using these alleles, we systematically investigated the role of endocytic trafficking in Dpp distribution and signaling. An overview of the results is summarized in S1 Table.

Endocytic trafficking has been proposed to regulate the extracellular Dpp gradient through transcytosis [19], recycling [20], or functioning as a sink [23–25]. We found that blocking Dynamin or Rab5 expands the extracellular Dpp distribution (Fig 4), whereas inhibition of downstream trafficking steps—including MVB formation (Fig 6), late endosome function (Fig 7), or recycling endosome formation (Fig 8)—has little to no effect. These results argue against transcytosis or recycling

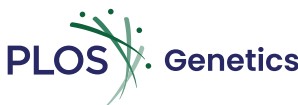

Fig 8. **Rab4 and Rab11 are largely dispensable for the formation of both the extracellular Dpp gradient and the Dpp signaling gradient.** (A-B) α-pMad staining (A), extracellular α-GFP staining in the lateral side (B), and extracellular α-GFP staining in the basal side (B') of $ap^{ts}$ >*rab4 RNAi* wing disc. (C-E) Average fluorescence intensity profiles of (A-B). Data are presented as mean+/- SD. (F-G) pMad staining (F), extracellular α-GFP staining

of the lateral side (G), and extracellular α-GFP staining of the basal side (G') of *ap^ts^>rab11 RNAi*. (H-J) Average fluorescence intensity profiles of (F-G). Data are presented as mean+/- SD. (K) mGL-Dpp (total) fluorescent signal of lateral side (K), with magnified region of the dorsal compartment (K') and the ventral compartment (K"). (L) Comparison of the number of mGL-Dpp puncta between K' and K". Two-sided Mann−Whitney test was used for the comparison (*p*=0.8413). (M-N) mGL-Dpp (total) fluorescent signal of the basal side (M), with magnified region of the dorsal compartment (M') and the ventral compartment (M") of *ap^ts^>rab11 RNAi* wing disc. (N) Comparison of the size of mGL-Dpp puncta between M' and M". Paired t-test was used for the comparison (*p*=0.0017).

as major contributors to Dpp spreading [19,20] and instead support a model in which endocytosis primarily functions as a sink to remove extracellular Dpp (Fig 9).

Given that loss of *tkv* also expands the extracellular Dpp distribution [23,24], Tkv-dependent internalization likely functions as a sink to restrict the range of Dpp (Fig 9). Interestingly, while loss of *shi* caused a broad, uniform expansion of the extracellular Dpp distribution, loss of Tkv primarily increased extracellular Dpp levels in central regions, while maintaining a graded distribution [24]. These observations suggest the existence of two distinct modes of Dpp internalization: a Tkv-dependent mechanism operating near the source that contributes to the Dpp signaling gradient, and a Tkv-independent mechanism acting in peripheral regions without contributing to signaling.

Upon internalization, Dpp is trafficked through distinct endocytic compartments (Fig 2). Since blocking Dynamin abolishes Dpp signaling, internalized Dpp is required for signaling activation. This suggests that the localization of Dpp within specific endocytic compartments is functionally linked to its signaling activity.

Loss of Rab5 resulted in increased Dpp signaling (Fig 3) accompanied by accumulation of Dpp and Tkv likely within early endocytic vesicles that fail to fuse with early endosomes, particularly on the basal side (Fig 4). Although the precise identity of these vesicular compartments remains unclear due to the absence of compartment-specific markers, partial degradation of Tkv was sufficient to suppress the enhanced Dpp signaling (Fig 5), suggesting that excess Tkv is required for the phenotype. We therefore propose that Dpp signaling is initiated in early endocytic vesicles prior to their fusion with early endosomes, and that signaling is subsequently terminated via Rab5-mediated trafficking (Fig 9). In contrast to our findings, earlier studies reported that loss of Rab5 reduces Dpp signaling [19,30]. We speculate that these opposing results may stem from pleiotropic effects. In our study, we knocked down Rab5 temporally using Gal80ts, allowing precise control of RNAi induction and minimizing developmental disruption. In contrast, the previous study used constitutive expression of dominant-negative Rab5 without Gal80ts [19]. Although RNAi was induced at lower temperatures to reduce potential artifacts, chronic low-level expression of the dominant-negative forms may have still led to broader cellular dysfunction that ultimately impairs Dpp signaling.

We next asked which endocytic trafficking compartment terminates Dpp signaling downstream of Rab5. While loss of ESCRT components led to an increase in Dpp signaling (Fig 6), loss of Rab7 did not (Fig 7). These results suggest that Dpp signaling is terminated at the level of MVBs rather than through lysosomal degradation of activated Tkv [49]. ESCRT components are involved in multiple cellular processes, including ILV formation [50], nuclear envelope reformation [51,52], and cytokinetic abscission [53]. Among these functions, we propose a model in which ESCRT machinery facilitates the sorting of ubiquitinated Tkv into ILVs, thus physically separating activated Tkv from its cytosolic effector Mad and terminating Dpp signaling (Fig 9). Notably, we observed that loss of the ESCRT complex results in a greater increase in Dpp signaling than loss of Rab5 (Figs 3D, 3G, 3H and 6A–6C). Since Rab5 loss impairs Dpp internalization (Fig 4C), whereas loss of the ESCRT complex does not disrupt endocytosis (Fig 6J and 6K), the differential effects on Dpp signaling likely reflect the reduced internalized Dpp available for signaling in Rab5 RNAi discs compared to ESCRT RNAi discs.

Interestingly, blocking MVB formation expanded the Dpp signaling gradient (Fig 6A–6C) and the intracellular Dpp distribution (Fig 6F–6I) without altering the extracellular Dpp gradient (Fig 6J and 6K). This suggests that the duration of intracellular Dpp signaling plays a critical role in shaping the signaling activity gradient derived from the extracellular Dpp gradient (Fig 9). Given that multiple signaling pathways are activated upon blocking MVB formation, it would be interesting

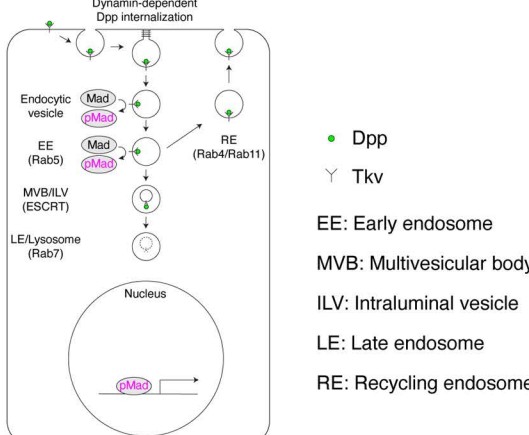

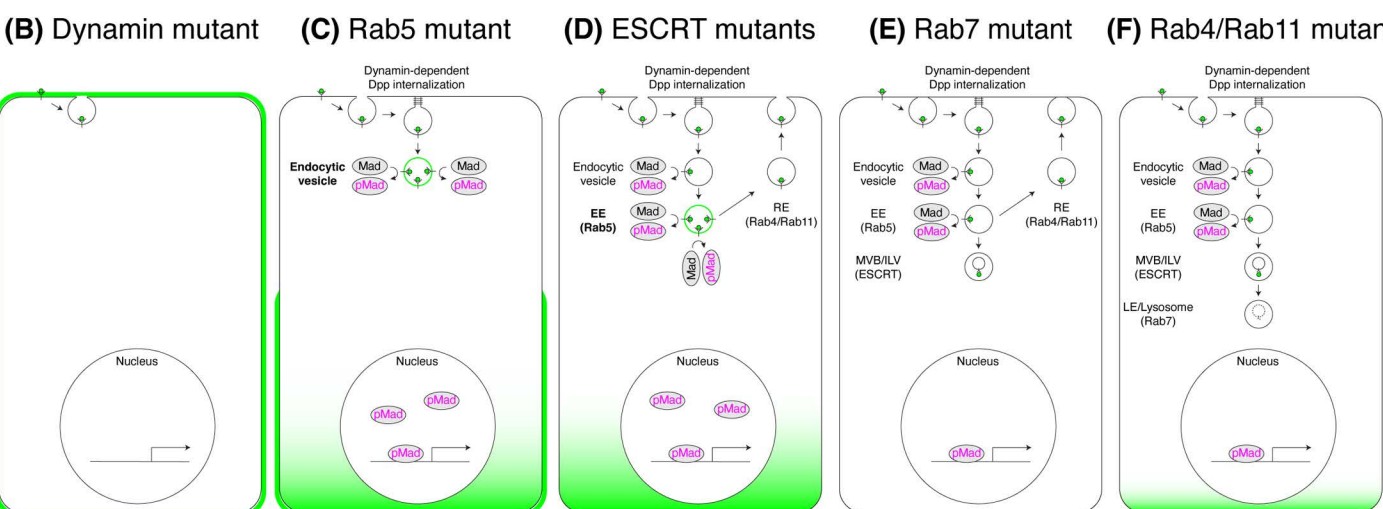

**Fig 9. A model for extracellular Dpp gradient formation and interpretation by endocytic trafficking.** (A) A model for extracellular Dpp gradient formation and interpretation by endocytic trafficking in wild type wing disc. Upon binding to its receptor Tkv, Dpp is internalized via Dynamin-dependent endocytosis. Dpp signaling is activated in endocytic vesicles before fusing with early endosomes (EE) and terminated at multivesicular bodies (MVBs), where activated Tkv is sorted into intraluminal vesicles (ILVs), physically separating Tkv from its downstream effector Mad. In the schematic model, Dpp is shown to be internalized from the apical side, although Dpp can be internalized from the basolateral side as well. (B) In Dynamin mutants (*shi*^ts), Dynamin-dependent Dpp internalization is impaired. As a result, Dpp accumulates on the cell surface and Dpp signaling is lost. (C) In Rab5 mutants, endocytosis, EE formation, and following endocytic trafficking are disrupted. Dpp accumulates on the cell surface and both Dpp and Tkv are retained in the endocytic vesicles, particularly at the basal side, leading to enhance Dpp signaling. (D) In ESCRT mutants, MVB formation and following endocytic trafficking are impaired. Although Dpp internalization proceeds normally, Dpp and Tkv accumulate in EE, resulting in enhanced Dpp signaling. (E) In Rab7 mutant, the formation of late endosomes (LEs) and subsequent lysosomal degradation are impaired. While Dpp internalization remains unaffected, Dpp and Tkv likely accumulate at MVBs without enhancing Dpp signaling. (F) In Rab4 and/or Rab7 mutant, recycling endosome (RE) formation is disrupted. Despite this, extracellular Dpp distribution and Dpp signaling are not significantly affected.

to test whether elevated signaling pathways result from impaired sorting of the activated receptors into the ILVs or lysosomal degradation.

By searching endocytic compartments where Dpp signaling is terminated, we found that loss of Rab7, Rab4, or Rab11 did not significantly alter Dpp distribution or signaling activity (Figs 7 and 8), despite the high degree of co-localization between Dpp and each of these Rab proteins (Fig 2). Since Rab7 is not required for MVB biogenesis [46], we speculate

that loss of Rab7 causes accumulation of Dpp and Tkv outside ILVs without enhancing Dpp signaling (Fig 9), although we were unable to determine the precise subcellular localization of Dpp and Tkv due to the lack of reliable fluorescent markers or antibodies for endogenous MVBs. Nevertheless, the overall distribution of Dpp and Tkv remained unchanged, possibly due to pH-dependent quenching of fluorophores within MVBs. Alternatively, Dpp and Tkv may still undergo degradation, as Rab7 deficiency has been reported to induce lysosome formation from recycling endosomes, thereby promoting degradation of cell surface proteins [54]. It also remains unclear how endocytic trafficking and Dpp distribution are regulated to maintain overall Dpp levels and signaling in the absence of Rab4 and/or Rab11. Notably, Rab11 knockdown caused accumulation of Dpp likely in enlarged early endosomes on the basal side (Fig 8M and 8N). However, this accumulation does not appear to enhance Dpp signaling, likely because Rab11 knockdown promotes subsequent trafficking to late endosomes and lysosomes [48], in contrast to the trafficking blockage caused by loss of the ESCRT complex.

### Novel *dpp* alleles to visualize endogenous Dpp morphogen gradient

The Dpp morphogen gradient has been intensively studied using GFP-Dpp. When overexpressed in the anterior stripe of cells—the main Dpp source—GFP-Dpp exhibits strong fluorescence in the source cells and a shallow, bilateral graded distribution [19,27]. Using FRAP and FCS, key parameters including the diffusion coefficient, degradation rate, and decay length have been measured [26,28]. However, the physiological relevance of these measurements has been questioned, as GFP-Dpp is typically overexpressed at levels approximately 400-fold higher than endogenous Dpp [20]. Indeed, in contrast to previous findings based on GFP-Dpp overexpression [20], we did not observe significant changes in extracellular Dpp distribution or signaling upon Rab4 and/or Rab11 knockdown (Figs 8 and S3), underscoring the potential discrepancies between overexpression systems and endogenous Dpp regulation.

Recent advances in genome engineering have enabled precise tagging of the endogenous *dpp* locus [8,10,20]. The *GFP-dpp* allele reveals that the endogenous fluorescent signal is too weak to visualize the graded Dpp distribution (Fig 1A) or to perform a FRAP assay [20]. Similarly, the *HA-dpp* allele showed a shallow extracellular HA-Dpp gradient, but conventional immunostaining failed to detect Dpp outside the primary source cells [10]. Although nanobody-based internalization assays allowed for visualization of internalized GFP-Dpp, it remains uncertain whether nanobody-bound GFP-Dpp accurately represents the functional ligand undergoing native trafficking.

The newly developed *mGL-dpp* and *mSC-dpp* alleles overcome these limitations. These alleles are functional during wing development and produce brighter fluorescence, enabling direct visualization of endogenous Dpp distribution—predominantly the internalized population—without requiring artificial overexpression or labeling (Fig 1). In addition, extracellular Dpp can be visualized using α-GFP antibody staining in the *mGL-dpp* background. FRAP and morphotrap have already been successfully applied to mGL-Dpp in Drosophila [55]. Applying these tools in the wing disc offers a powerful opportunity to study the dynamics of Dpp gradient formation under physiological conditions.

## Materials and methods

### Fly stocks

Flies for experiments were kept in standard fly vials containing polenta and yeast. Embryos from fly crosses for experiments including Gal80ts were collected for 24h and kept at 18°C, until shifted to 29°C prior to dissection of 3rd instar larvae. To induce *Rab5²* clones, third instar larvae were subjected to heat shock (37°C) for 8 minute and incubated at 25°C for 24 hours prior to dissection. The following fly lines were used: *shibire^{ts1}* (BDSC 7068), *mGL-dpp* (this study), *mSC-dpp* (this study), *ap-Gal4* (Bloomington 3041), *ap[c1.4b-Gal4]* (Michèle Sickmann and Martin Müller), *tub-GAL80TS* (M. Affolter), *tkv-3xHA* (G. Pyrowolakis), *tkv-YFP* (G. Pyrowolakis), *tkv-1xHAeGFP* (G. Pyrowolakis), *brk^{XA}* (G. Campbell & A. Tomlinson), *UAS-rab5-RNAi* (BDSC 30518, VDRC 34096, 103945), *UAS-rab5.S43N* (BDSC 42703 & 42704), *UAS-rab4 RNAi* (VDRC 24672), *UAS-rab11-RNAi* (VDRC 22198), *UAS-vps4-RNAi* (VDRC 105977), *UAS-tsg101-RNAi* (BDSC 35710), *UAS-shrub-RNAi* (BDSC 38305), *UAS-rab7-RNAi* (BDSC 27051), *dpp-LacZ* (M.Affolter), *UAS-LOT-deGradHA* (G.

Pyrowolakis & M. Affolter), *rab5-eYFP* (BDSC 62543*), rab7-eYFP* (BDSC 62545), *rab4-eYFP* (BDSC 62542*), rab11-eYFP* (BDSC 62549), *FRT82b, rab7*^Gal4-Knock-in null allele (P. R. Hiesinger), *hsFlp,UAS-GFP,w;FRT42D,tub-Gal80;tub-Gal4, FRT82B,tub-Gal80* (BDSC 86318), *hsFlp;tub > CD2 > Gal4,UAS-lacZ* (B. Bello), *hsFlp, rab5², FRT40* (BDSC 42702), *yw, dpp^d8* and *dpp^d12* are described from Flybase.

## Genotypes by figures

All fly genotypes used in this study are listed in Table 1.

## Generation of *mGL-dpp* and *mSC-dpp*

The detail procedure to generate endogenously tagged *dpp* alleles were previously reported [10]. In brief, utilizing the attP sites in a MiMIC transposon inserted in the dpp locus (MiMIC dppMI03752, BDSC 36399), about 4.4 kb of the dpp genomic sequences containing the second (last) coding exon of dpp including a tag and its flanking sequences was inserted in the intron between dpp's two coding exons. The endogenous exon was then removed using FLP-FRT to keep only the tagged exon. mGL (mGreenLantern [31]) was inserted after the last processing site to tag all the Dpp mature ligands. mGL-dpp homozygous flies show no obvious phenotypes.

## Immunohistochemistry

**Visualization of mGL-Dpp and mSC-Dpp.** To visualize the (total) mGL-Dpp and mSC-Dpp signal, third instar larvae were dissected in ice-cold Phosphate Buffered Saline (PBS). The dissected larvae were washed with HCl with pH 3.0 following the acid wash protocol [20] to remove the extracellular proteins prior to fixation in 4.0% Paraformaldehyde (PFA) for 25min on a shaker at room temperature (25°C). The discs were washed three times for ten minutes with PBS at 4°C and mounted in Vectashield on glass slides.

**Total staining.** Third instar larvae were dissected in ice-cold Phosphate Buffered Saline (PBS) and fixed in 4.0% Paraformaldehyde (PFA) for 25min on a shaker at room temperature (25°C). After fixation, the discs were washed three times for ten minutes with PBS at 4°C, and three times with PBST (0.3% Triton-X in PBS) to permeabilize the tissues. The discs were then blocked in 5% normal goat serum (NGS) in PBST for 30min. The primary antibodies were added to 5% NGS in PBST for incubation of the discs at 4°C overnight. The next day, the primary antibody was carefully removed, and the samples were washed three times with PBST. Secondary antibodies were added to 5% NGS in PBST and the discs were incubated for 2h in the dark at room temperature. At last, the samples were washed three times for 15 minutes with PBST at room temperature, two times with PBS, and mounted in Vectashield on glass slides.

**Extracellular staining.** Wing discs from third instar larvae were dissected in ice-cold Schneider's Drosophila medium (S2). The discs were then blocked in cold 5% NGS in S2 medium on ice for 10min. The blocking solution was carefully removed, and the primary antibody was added for 1h on ice. To ensure an even distribution of the primary antibody, the tubes were tapped every 10min during the incubation time. The antibody was then removed, and the samples were washed at least 6 times with cold S2 medium and another two times with cold PBS to remove excess primary antibody. Wing discs were then fixed with 4% PFA in PBS for 25min on the shaker at room temperature (25°C). After fixation the protocol continued as described in total staining.

**Acid wash.** The protocol was adapted from [20]. To remove the extracellular proteins prior to fixation, the dissected wing discs were washed three times ten seconds with ice-cold Schneider's Drosophila medium (S2), pH dropped down to 3 by HCl. To remove the stripped membrane-bound proteins, the discs were washed three times 15min with ice-cold S2 medium (pH 7.4) and fixed in 4% PFA.

**Antibodies.** Primary antibodies: Rabbit anti-phospho-Smad 1/5 (Cell signaling 9516S; 1:200), mouse anti-patched (DSHB; 1:40), mouse anti-wingless (4D4, DSHB; 1:120), rabbit anti-GFP (Abcam ab6556; 1:2000 for total staining, 1:200 for extracellular staining,), guinea pig anti-rab5 (provided by Akira Nakamura; 1:1000), rabbit anti-rab11 (provided by Akira

**Table 1. List of genotypes used in this study.**

| |
|---|
| <u>Fig 1A</u>; *GFP-dpp/Cyo, p23* |
| <u>Fig 1B</u>; *mGL-dpp/+* |
| <u>Fig 1C</u>; *mSC-dpp/+* |
| <u>Fig 1D</u>; *yw* |
| <u>Fig 1E</u>; *JAX; mGL-dpp/mGL-dpp* |
| <u>Fig 1F</u>; *JAX; mSC-dpp/mSC-dpp* |
| <u>Fig 1G</u>; *HA-dpp/HA-dpp* |
| <u>Fig 1H</u>; *JAX; HA-dpp/HA-dpp* |
| <u>Fig 1I</u>; *JAX; mGL-dpp/mGL-dpp* |
| <u>Fig 1J</u>; *JAX; mSC-dpp/mSC-dpp* |
| <u>Fig 1L</u>; *mGL-dpp/+* |
| <u>Fig 2A</u>; *mSC-dpp/ rab5-eYFP* |
| <u>Fig 2B</u>; *mSC-dpp/ +; rab7-eYFP/ +* |
| <u>Fig 2C</u>; *mSC-dpp/ rab4-eYFP* |
| <u>Fig 2D</u>; *mSC-dpp/ +; rab11-eYFP/ +* |
| <u>Fig 3A</u>; *shi^{ts}/+ (2h heat shock at 34°C)* |
| <u>Fig 3B</u>; *shi^{ts} (2h heat shock at 34°C)* |
| <u>Fig 3D</u> and <u>3E</u>; *HA-dpp, ap-Gal4/ +; UAS-rab5-RNAi (30518)/ tub-Gal80ts (29h at 29°C)* |
| <u>Fig 3G</u>; *HA-dpp, ap-Gal4/ +; UAS-rab5-RNAi (34096)/ tub-Gal80ts (29h at 29°C)* |
| <u>Fig 3H</u>; *HA-dpp, ap-Gal4/ +; UAS-rab5-RNAi (103945)/ tub-Gal80ts (24h at 29°C)* |
| <u>Fig 3I</u>; *HA-dpp, ap-Gal4/ UAS-rab5.S43N (42703); tub-Gal80ts/ + (18h at 29°C)* |
| <u>Fig 3J</u>; *Ollas-dpp, ap[c1.4b-Gal4]/ +; UAS-rab5.S43N (42704)/ tub-Gal80ts (13.5h at 29°C)* |
| <u>Fig 3K</u>; *hsFlp, rab5^2 FRT40/arm-LacZ, m(2)Z FRT40* |
| <u>Fig 4A</u>; *brk^{XA}; dpp^{d8}, ap-Gal4/ dpp^{d12}; UAS-rab5-RNAi (30518)/ tub-Gal80ts (29h at 29°C)* |
| <u>Fig 4B</u>; *ap-Gal4/ dpp-LacZ; UAS-rab5-RNAi (30518)/ tub-Gal80ts (29h at 29°C)* |
| <u>Fig 4C</u> and <u>4D</u>; *mGL-dpp, ap[c1.4b-Gal4]/ +; UAS-rab5-RNAi (30518)/ tub-Gal80ts (29h at 29°C)* |
| <u>Fig 4F</u>; *shi^{ts}/ +; mGL-dpp/+ (2h heat shock at 34°C)* |
| <u>Fig 4G</u>; *shi^{ts}; mGL-dpp/+ (2h heat shock at 34°C)* |
| <u>Fig 4I</u> and <u>4J</u>; *mGL-dpp, ap[c1.4b-Gal4]/ +; UAS-rab5-RNAi (30518)/ tub-Gal80ts (29h at 29°C)* |
| <u>Fig 4K</u> and <u>4L</u>; *tkv-YFP, ap[c1.4b-Gal4]; UAS-rab5-RNAi (30518)/ tub-Gal80ts (29h at 29°C)* |
| <u>Fig 5A</u>; *yw, tkv-HA-eGFP, ap-Gal4/ +; tub-Gal80ts/ + (29h at 29°C)* |
| <u>Fig 5B</u>; *tkv-HA-eGFP, ap-Gal4/ +; UAS-rab5-RNAi (30518)/ tub-Gal80ts (29h at 29°C)* |
| <u>Fig 5C</u>; *tkv-HA-eGFP, ap-Gal4/ +; UAS-rab5-RNAi (30518), tub-Gal80ts/ UAS-deGradHA (29h at 29°C)* |
| <u>Fig 5D</u>; *tkv-HA-eGFP, ap-Gal4/ +; UAS-deGradHA/ tub-Gal80ts (29h at 29°C)* |
| <u>Fig 6A</u>; *tkv-YFP, ap[c1.4b-Gal4]; UAS-tsg101-RNAi (35710)/ tub-Gal80ts (44h at 29°C)* |
| <u>Fig 6B</u>; *tkv-YFP, ap[c1.4b-Gal4]/ UAS-shrub-RNAi (38305); tub-Gal80ts/ + (28h at 29°C)* |
| <u>Fig 6C</u> and <u>6D</u>; *tkv-YFP, ap[c1.4b-Gal4]/ UAS-vps4-RNAi (105977); tub-Gal80ts/ + (30h at 29°C)* |
| <u>Fig 6E</u>; *tkv-YFP, ap[c1.4b-Gal4]; UAS-rab7-RNAi (27051)/ tub-Gal80ts (42h at 29°C)* |
| <u>Fig 6F</u> and <u>6J</u>; *mGL-dpp, ap[c1.4b-Gal4]/ UAS-vps4-RNAi (105977); tub-Gal80ts/ + (30h at 29°C)* |
| <u>Fig 6H</u>; *mGL-dpp, ap[c1.4b-Gal4]/ rab5-eYFP; UAS-tsg101-RNAi (35710)/ tub-Gal80ts (44h at 29°C)* |
| <u>Fig 7A</u>; *hsFlp, UAS-GFP; FRT82b, tub-Gal4/ FRT82b rab7 Gal4-Knock-In* |
| <u>Fig 7B</u> and <u>7C</u>; *HA-dpp, ap-Gal4; UAS-rab7-RNAi (27051)/ tub-Gal80ts (42h at 29°C)* |
| <u>Fig 7E</u> and <u>7G</u>; *mGL-dpp, ap[c1.4b-Gal4]; UAS-rab7-RNAi (27051)/ tub-Gal80ts (42h at 29°C)* |
| <u>Fig 8</u>, <u>8B</u> and <u>8K</u>; *mGL-dpp, ap[c1.4b-Gal4]/ +; UAS-rab4-RNAi (24672)/ tub-Gal80ts (42h at 29°C)* |
| <u>Fig 8F</u>, <u>8G</u> and <u>8M</u>; *mGL-dpp, ap[c1.4b-Gal4]/ +; UAS-rab11-RNAi (22198)/ tub-Gal80ts (42h at 29°C)* |
| <u>S1A</u> and <u>S1B Fig</u>; *JAX; mGL-dpp/mGL-dpp.* |
| <u>S2A Fig</u>; *tkv-YFP, ap[c1.4b-Gal4]/ +; UAS-tsg101-RNAi (35710)/ tub-Gal80ts (44h at 29°C)* |

*(Continued)*

**Table 1.** (Continued)

| |
|---|
| Fig 1A; *GFP-dpp/Cyo, p23* |
| S2B Fig; *tkv-YFP, ap[c1.4b-Gal4]/ UAS-shrub-RNAi (38305); tub-Gal80ts/ +* (28h at 29°C) |
| S3A and S3B Fig; *mGL-dpp, ap[c1.4b-Gal4]/;UAS-rab4-RNAi (24672),UAS-rab11-RNAi (22198)/ tub-Gal80ts* (42h at 29°C). |

Nakamura; 1:8000), mouse anti-rab7 (DSHB; 1:30), mouse anti-ubiquitin (Enzo PW8810–0100; 1:1000), mouse anti-beta galactosidase (Promega Z378825580610; 1:500), guinea pig anti-brk (provided by from Gines Morata; 1:1000), mouse anti-V5 (Invitrogen; 1:5000).

The following secondary antibodies were used at 1:500 dilutions in this study: Goat anti-rabbit IgG (H+L) Alexa Fluor 488 (A11008 Thermo Fischer), goat-anti-rabbit IgG (H+L) Alexa Fluor 568 (A11011 Thermo Fischer), goat-anti-rabbit IgG (H+L) Alexa Fluor 680 (A21109 Thermo Fischer), goat anti-mouse IgG (H+L) Alexa Fluor 488 (A11001 Thermo Fischer), goat anti-mouse IgG (H+L) Alexa Fluor 568 (A11004 Thermo Fischer), goat anti-mouse IgG (H+L) Alexa Fluor 680 (A10038 Thermo Fischer), goat-anti-guinea pig IgG (H+L) Alexa Fluor 568 (A11075 Thermo Fischer), goat-anti-guinea pig IgG (H+L) DyLight 680 (SA5–10098 Invitrogen).

**Imaging.** Wing imaginal discs were imaged using a Leica SP5-II MATRIX and an Olympus Spinning Disk (CSU-W1), and images were analyzed using Fiji (ImageJ). Figs 1–9 were obtained using OMERO and Adobe Illustrator.

**Quantification of pMad and extracellular mGL-dpp intensity.** To quantify the intensity of pMad and extracellular mGL-dpp gradient in the images, an average intensity of three sequential stacks was created using Fiji ImageJ (v1.53c). Each signal intensity profile collected in Excel (Ver. 16.51) was aligned along A/P compartment boundary (based on anti-Ptc or pMad staining) and average signal intensity profile from different samples was generated and plotted by the script (wing_disc-alignment.py). The average intensity of the samples and the control were then compared using the script (wingdisc_comparison.py). Both scripts were generated by E. Schmelzer and can be found on: https://etiennees.github.io/Wing_disc-alignment/. The resulting signal intensity profiles (mean with SD) were generated on GraphPad Prism software (v.9.3.1(471)). Figs 1–9 were prepared using OMERO (ver5.9.1) and Adobe Illustrator (24.1.3).

**Quantification of mGL-dpp and Tkv-YFP positive puncta.** To measure the number particles an average intensity of 3 z-stacks from the images were created using Fiji ImageJ. The total area of controls and samples in which the particles were counted had a width of 20.16 and height of 34.17 microns. The number and area of the particles were measured by the built-in "Analyze Particles" plug-in in Fiji. The data were used to make the graphs on GraphPad Prism. A ratio-paired t-test (p<0.05) was used for statistical analysis.

**Reproducibility.** All experiments were independently repeated at least two time, with consistent results. Statistical significance was assessed by the GraphPad Prism software (v.9.3.1(471)).

## Supporting information

**S1 Fig. Acid wash removes the extracellular mGL-Dpp without affecting the intracellular mGL-Dpp distribution.** (A-A') Extracellular α-GFP staining (A) and mGL-Dpp fluorescent signal (A') of control wing disc without acid wash. (B-B') Extracellular α-GFP staining (B) and mGL-Dpp fluorescent signal (B') after the acid wash. (C-C') Quantification of the extracellular GFP intensity of A and B (C), and the intracellular mGL fluorescent intensity in A' and B' (C'). (TIF)

**S2 Fig. Knockdown of the ESCRT components TSG101 and Shrub leads to an accumulation of Tkv and Ubiquitin in puncta.** (A-A") Tkv-YFP fluorescent signal (A), α-Ubiquitin staining (A'), and the merged image (A") of *ap^ts >tsg101 RNAi* wing disc. (B-B") Tkv-YFP fluorescent signal (B), α-Ubiquitin (B'), and the merged image (B") of *ap^ts>shrub RNAi* wing disc. (TIF)

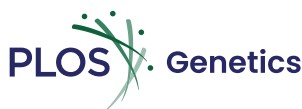

**S3 Fig. Simultaneous knocking down of Rab4 and Rab11 does not affect Dpp signaling gradient or extracellular mGL-Dpp gradient.** (A, B) α-pMad staining (A) and extracellular α-GFP staining (B) of *ap^ts^ > rab11 RNAi* wing disc. (C) Average fluorescence intensity profiles of (A). Data are presented as mean +/- SD. (D) Average fluorescence intensity profiles of (B). Data are presented as mean +/- SD. Scale bar: 30μm.
(TIF)

**S1 Table. The effects on extracellular/intracellular and basal/lateral localization of Dpp and Tkv as well as Dpp signalling for each genetic manipulation.**
(XLSX)

**S1 Data. Raw data to generate graphs and statistic analyses in this study.**
(XLSX)

## Acknowledgments

The authors would like to thank Markus Affolter for his continuous support throughout the course of this project. We thank Developmental Studies Hybridoma Bank (DSHB) at The University of Iowa for providing us with the primary antibodies, and Bloomington Drosophila Stock Center (BDSC) for providing us with fly stocks. We would also like to thank Giorgos Pyrowolakis, Peter Robin Hiesinger and Isabel Guerrero for providing us with fly lines and Akira Nakamura for providing us with primary antibodies. We thank Etienne Schmelzer for providing us with scripts for quantifications. We would like to thank Bernadette Bruno, Gina Evora, Karin Mauro and Dario Dörig for their constant and reliable supply of the world's best fly food. We thank the Biozentrum Imaging Core Facility (IMCF), especially Oliver Biehlmaier, Alexia Loyton-Ferrand, Sara Roig, Kai Schleicher, Laurent Guerard, Nikolaus Ehrenfeuchter and Sébastien Herbert for their constant support with the microscopes and image analysis.

We used OpenAI's ChatGPT to assist with language editing and improving the clarity of the manuscript.

## Author contributions

**Conceptualization:** Shinya Matsuda.

**Data curation:** Sheida Hadji Rasouliha.

**Formal analysis:** Sheida Hadji Rasouliha, Shinya Matsuda.

**Funding acquisition:** Shinya Matsuda.

**Investigation:** Sheida Hadji Rasouliha, Shinya Matsuda.

**Resources:** Gustavo Aguilar, Cindy Reinger, Shinya Matsuda.

**Supervision:** Shinya Matsuda.

**Writing – original draft:** Shinya Matsuda.

**Writing – review & editing:** Shinya Matsuda.

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
