## [Decision Letter · Decision Letter 0]

Response to Reviewers
Revised Manuscript with Track Changes
Manuscript

Aimée Dudley

Editor-in-Chief

PLOS Genetics

Anne Goriely

Editor-in-Chief

PLOS Genetics

**Additional Editor Comments (if provided):**
**Journal Requirements:**

At this stage, the following Authors/Authors require contributions: Sheida Hadji Rasouliha, Gustavo Aguilar, Cindy Reinger, and Shinya Matsuda. Please ensure that the full contributions of each author are acknowledged in the "Add/Edit/Remove Authors" section of our submission form.

The list of CRediT author contributions may be found here: https://journals.plos.org/plosgenetics/s/authorship#loc-author-contributions

3) Please provide an Author Summary. This should appear in your manuscript between the Abstract (if applicable) and the Introduction, and should be 150u2013200 words long. The aim should be to make your findings accessible to a wide audience that includes both scientists and non-scientists. Sample summaries can be found on our website under Submission Guidelines:

https://journals.plos.org/plosgenetics/s/submission-guidelines#loc-parts-of-a-submission

4) We do not publish any copyright or trademark symbols that usually accompany proprietary names, eg ©,  ®, or TM  (e.g. next to drug or reagent names). Therefore please remove all instances of trademark/copyright symbols throughout the text, including:

- TM on page: 24.

5) Please upload all main figures as separate Figure files in .tif or .eps format. For more information about how to convert and format your figure files please see our guidelines:

6) We notice that your supplementary Figures are included in the manuscript file. Please remove them and upload them with the file type 'Supporting Information'. Please ensure that each Supporting Information file has a legend listed in the manuscript after the references list.

7) We note that your Data Availability Statement is currently as follows: "All relevant data are within the manuscript and its Supporting Information files.". Please confirm at this time whether or not your submission contains all raw data required to replicate the results of your study. Authors must share the “minimal data set” for their submission. PLOS defines the minimal data set to consist of the data required to replicate all study findings reported in the article, as well as related metadata and methods (https://journals.plos.org/plosone/s/data-availability#loc-minimal-data-set-definition).

8) Please amend your detailed Financial Disclosure statement. This is published with the article. It must therefore be completed in full sentences and contain the exact wording you wish to be published.

1) State what role the funders took in the study. If the funders had no role in your study, please state: "The funders had no role in study design, data collection and analysis, decision to publish, or preparation of the manuscript.".

9) Your current Financial Disclosure states, "G.A. was supported by “Fellowships for Excellence” from the International PhD Program in Molecular Life Sciences of the Biozentrum, University of Basel. S.M. was supported by anSNSF Ambizione grant (PZ00P3_180019). ".

However, your funding information on the submission form indicates a different order for your funding sources, The Swiss National Science Foundation (SNSF) then “Fellowships for Excellence” from the International PhD Program in Molecular Life Sciences of the Biozentrum, University of Basel..

Please indicate by return email the full and correct funding information for your study and confirm the order in which funding contributions should appear. Please be sure to indicate whether the funders played any role in the study design, data collection and analysis, decision to publish, or preparation of the manuscript.

**Reviewers' comments:**

At this stage, the following Authors/Authors require contributions: Sheida Hadji Rasouliha, Gustavo Aguilar, Cindy Reinger, and Shinya Matsuda. Please ensure that the full contributions of each author are acknowledged in the "Add/Edit/Remove Authors" section of our submission form.

The list of CRediT author contributions may be found here: https://journals.plos.org/plosgenetics/s/authorship#loc-author-contributions

3) Please provide an Author Summary. This should appear in your manuscript between the Abstract (if applicable) and the Introduction, and should be 150u2013200 words long. The aim should be to make your findings accessible to a wide audience that includes both scientists and non-scientists. Sample summaries can be found on our website under Submission Guidelines:

https://journals.plos.org/plosgenetics/s/submission-guidelines#loc-parts-of-a-submission

4) We do not publish any copyright or trademark symbols that usually accompany proprietary names, eg ©,  ®, or TM  (e.g. next to drug or reagent names). Therefore please remove all instances of trademark/copyright symbols throughout the text, including:

- TM on page: 24.

5) Please upload all main figures as separate Figure files in .tif or .eps format. For more information about how to convert and format your figure files please see our guidelines:

6) We notice that your supplementary Figures are included in the manuscript file. Please remove them and upload them with the file type 'Supporting Information'. Please ensure that each Supporting Information file has a legend listed in the manuscript after the references list.

7) We note that your Data Availability Statement is currently as follows: "All relevant data are within the manuscript and its Supporting Information files.". Please confirm at this time whether or not your submission contains all raw data required to replicate the results of your study. Authors must share the “minimal data set” for their submission. PLOS defines the minimal data set to consist of the data required to replicate all study findings reported in the article, as well as related metadata and methods (https://journals.plos.org/plosone/s/data-availability#loc-minimal-data-set-definition).

8) Please amend your detailed Financial Disclosure statement. This is published with the article. It must therefore be completed in full sentences and contain the exact wording you wish to be published.

1) State what role the funders took in the study. If the funders had no role in your study, please state: "The funders had no role in study design, data collection and analysis, decision to publish, or preparation of the manuscript.".

9) Your current Financial Disclosure states, "G.A. was supported by “Fellowships for Excellence” from the International PhD Program in Molecular Life Sciences of the Biozentrum, University of Basel. S.M. was supported by anSNSF Ambizione grant (PZ00P3_180019). ".

However, your funding information on the submission form indicates a different order for your funding sources, The Swiss National Science Foundation (SNSF) then “Fellowships for Excellence” from the International PhD Program in Molecular Life Sciences of the Biozentrum, University of Basel..

Please indicate by return email the full and correct funding information for your study and confirm the order in which funding contributions should appear. Please be sure to indicate whether the funders played any role in the study design, data collection and analysis, decision to publish, or preparation of the manuscript.

**Comments to the Authors:**

Reviewer #1: This interesting manuscript considers a long-standing problem in the field of morphogen signalling, in particular relating to the mechanisms of DPP signalling, a key BMP morphogen, in Drosophila wing patterning. It relies on critical new tools, two lines of flies in which the endogenous DPP protein is tagged with green and red fluorescent proteins. Unlike previous endogenous fluorescently tagged Dpp lines, it is possible to visualise intracellular and extracellular DPP in wing discs from these lines and therefore assess the effects of different genetic manipulations on the endogenous protein, and this leads to some findings that contrast with experiments undertaken with overexpressed DPP.

Using these tools and some genetic tricks to allow the genetically manipulated animals to consistently survive through larval development, the authors provide evidence that Dynamin-mediated internalisation is required to activate DPP signalling. However, early endosomal Rab5 is not required either for signal activation or signal spread, though it does play a role in downregulating DPP signalling, probably through ESCRT-mediated internalisation of TKV-containing ILVs in MVBs. Rab7 knockdown experiments suggest that late endosome formation is not essential for this process.

Overall, I thought the experiments presented in the manuscript were generally of a high quality. The interpretation is inevitably limited by the fact that many of the genetic manipulations will have knock-on cell biological effects in knockdown/mutant cells, which might suppress or mask the direct effects of the manipulations. But importantly, there are differences in the effects of the different manipulations, which suggest that only some of the steps tested have a role in DPP signalling activation and signal downregulation.

I have a few suggestions that I think would help to support the conclusions made or aid clarity.

1. Figure 4: I wondered whether the authors had tried to determine the identity of basal compartments containing more DPP/TKV in Rab5 knockdown cells by using YFP-Rab markers and the red-DPP they employ. It is not clear to me where endocytosed DPP will reside if it is not trafficked to the early endosome.

2. Figure 6: More importantly, I think some more analysis is required to interpret the roles of ESCRTs versus Rab7. There was no analysis of co-localisation of Ubi and Tkv in ESCRT knockdowns - were most of the large Tkv-containing compartments Ubi-positive or just a small fraction? Knowing this might allow us to determine whether these are mainly stalled MVB compartments. And since DPP accumulates in large compartments in ESCRT knockdown cells, could the Rab identity of these compartments be analysed with the red-DPP marker to see whether the DPP resides in late or recycling endosomal compartments? Figure 6H’’’ is also missing, so the important data quantification is absent. Finally, in the Rab7 knockdown cells, levels of TKV seem largely unaffected. But if late endosomes and lysosomes are not forming, wouldn’t TKV become trapped in accumulating MVBs? This point needs some consideration in the Discussion.

3. Figure 7: Figure 2 suggests there is quite strong co-localisation between DPP and Rab7. So if Rab7 is knocked down, where does that DPP now reside? Again, this could be assessed with the red-DPP and labelled Rabs. Regarding the Rab7 experiments in general, I believe it would also be useful to quote one or more previous studies that suggest Rab7 is not required for MVB biogenesis, eg. Rab7 after, eg: https://pubmed.ncbi.nlm.nih.gov/19265192/.

4. I think the authors should include a table of the effects on extracellular/intracellular, basal/lateral DPP localisation, Tkv and DPP signalling for each genetic manipulation either with Figure 9 or in a Supplementary Figure. I think this would help to contrast the different data presented.

5. For Figure 9, I also suggest the authors include the different genes tested at different stages of the endocytic pathway on the diagram (and potentially add in Rab4 and Rab11 recycling steps, so it is clear where they intersect with the degradative pathway).

6. In the Discussion, there should at least be some reference to the fact that ESCRTs are not just involved in ILV formation, even though the interpretation that ILV blockade is how they likely function in DPP signalling downregulation seems reasonable.

7. In the Discussion, it is suggested that the data ‘support the idea that early endocytosis simply acts as a sink for Dpp’. I think that needs some more clarity, with ‘early endocytosis’ replaced by ‘trafficking to the early endosome’. The Rab11 knockdown result suggests that endosomal and DPP trafficking is significantly disrupted in knockdown cells, but DPP signalling is unaffected. Do the authors think that DPP is accumulating in the early endosomal sink here?

Minor points

1. Figure 1: I wondered whether the authors could be certain that the GFP antibody employed does not affect GFP fluorescence by referring to previous studies using this antibody?

2. Statistics – all the statistics use a t-test, which assumes parametrically distributed data points. Has any test been undertaken to confirm that this is the case (this should be mentioned), therefore showing that a non-parametric test is not required?

3. Although generally well-written, there are a number of minor typos throughout the text, eg.

Page 3, near bottom of last paragraph: ‘the duration of Dpp signalling affects interpretation of the extracellular …’

Page 5, bottom title – ‘Rab5-mediated’

Figure 1 legend: ‘anti-GFP staining’ or ‘α-GFP’, ‘50 µm’

Reviewer #2: This manuscript by Rasouliha and colleagues examines the cellular mechanisms controlling Dpp signaling gradients in the Drosophila fly wing using molecular genetic and cellular approaches. A major strength of the study is that the cell biological experiments are done in an in vivo (albeit dissected) context, which adds considerable value (and difficulty) to the work. Some of this was facilitated by the generation of endogenously tagged alleles of Dpp, which may be useful to the field. Another strength and point of interest is the focus on cell signaling and morphogen gradients, including data that contradicts some of what has been previously reported for mechanisms controlling Dpp signaling and gradients. In the end, I found the central model, that most Dpp signaling (pMad activation) occurs intracellularly following Dpp internalization but before quenching in MVBs, to be convincing. Perhaps less clear was how intracellular trafficking pathways impact extracellular Dpp accumulation or why ESCRT seems to affect Dpp signaling differently than rab5, although these are more minor points.

Major issue:

In general, this paper needs substantial grammatical/English editing throughout to improve clarity, readability, and correct errors including some typos. In the end I was able to work through most of the sections and gain a sufficient understanding of what the authors meant to convey, which allowed me to review the science. Still, there were numerous sentences that I ultimately remained confused about. As such, the writing needs considerable work, preferably with the aid of a professional copy editor or equivalent. Doing this prior to submitting is preferrable. I also thought that several sections would benefit from more substantive background information and explanations, which many readers would require regardless of grammar. To me the writing was overly terse.

Below are specific points relevant to each section of the manuscript for the authors to consider.

1) Figure 1/S1 and associated text.

i) Non-fly readers may benefit from an example of WT to accompany D/E.

ii) Here and throughout, certain terms and descriptions were somewhat confusing. For example, “Dpp signal” means pMad (phosphorylation) whereas “mGL-Dpp signal” means the fluorescently tagged Dpp protein.

iii) State that anti-GFP antibodies recognize mGL, from which it is derived.

iv) I was somewhat confused by the anti-GFP and mGL co-localization study. Lack of co-loc or apparent correlation in these expression patterns seems surprising since they are looking at the same molecule, even if one shows both intra and extracellular expression and the other extracellular only. In addition, the gradients of extracellular Dpp seemed very shallow (visually) relative to pMad and intracellular Dpp. Some explanations here seem warranted.

v) S1A and 1K’ look rather different although they are both anti-GFP of mGL-Dpp. The somewhat graded expression in S1A, with higher levels in the center, also correlates more mGL, which is what one might expect. I wasn’t sure what to make of this.

2) Figure 2 and associated text.

i) Might want to clarify that the Mander’s overlap adds up to ~150% probably because of some overlap in the trafficking markers or just coincidental co-loc.

ii) Not here, but later it becomes somewhat confusing that recycling doesn’t seem to play a role in Dpp expression despite the observed co-loc to rab4 and rab11 compartments.

3) Figure 3 and associated text.

i) It might be useful to provide more complete genotypes in the figure legends that would include gal80, which was specifically mentioned in the text.

ii) Please indicate the dorsal compartment for at least one of the panels. This would be useful in several other figures as well. Many readers won’t know what they are looking at or where the ap driver is active.

4) Figure 4 and associated text.

i) I was initially confused by the wording introducing the experiment shown in 4A. Again, given that this is a general audience and that there are no word limits, it would be helpful to explain some of the experiments more fully.

ii) I wasn’t given sufficient information or background to understand the meaning of the results in 4I-P or to know why the specific conclusion was made.

iii) Looking at the images in K and L I had a hard time seeing discrete puncta and thus wondered about the quantification using # of puncta in O and P.

5) Figure 5 and associated text.

i) I was a bit puzzled as to why the dorsal expression of pMad was relatively unaffected in the WT background versus the rab5 rnai. To my eye, the level of TKv-eGFP in the dorsal half of the 5C disc was clearly lower than the ventral side. But that was less the case for 5D. If knockdown was less in 5D for some reason, that could explain this discrepancy.

ii) The final concluding sentence, “These results suggest that Dpp signaling is not terminated in the absence of Rab5 due to impaired downregulation of activated Tkv.”, was confusing to me. This is one example of my major point above. Dpp signaling is shown to be increased in the absence of rab5, so the use of “terminated” was puzzling in this context. To me, these results show that increased Dpp pathway activation following rab5 inhibition is dependent on the Dpp receptor, Tkv. Is there something that I’m missing? Regardless, the sentence did not clearly summarize the result or its interpretation.

6) Figure 6/S2 and associated text.

i) “We then asked in which endocytic trafficking Dpp signal is terminated?” Another example of lack of clarity and grammatical issues. A corrected sentence might read, “We next asked in which endocytic trafficking compartment Dpp signaling is terminated.” Please have this manuscript edited.

ii) Although the results were generally quite clear from the images, I might have expected somewhat more quantification of the data (e.g., the difference between H’ and H”).

iii) Is the idea that Dpp/receptors are ubiquitylated or that inhibiting cargo sorting into MVBs leads to the general accumulation of ubiquitylated proteins in cells?

iv) It appears that the effect of ESCRT rnai on pMad greater than that observed for rab5 rnai? If so, why would that be?

7) Figure 7 and associated text.

i) What is the explanation for the lack of any difference in mGL-Dpp when rab7 is inhibited? Has endosomal Dpp been degraded despite rab7 knockdown? Has the pH dropped sufficiently within endosomes/MVBs to quench the mGL signal?

8) Figure 8 and associated text.

i) Is it possible that Dpp can be recycled by either slow (rab11) or fast (rab4) recycling pathways and that both would have to be compromised before a strong change in the Dpp gradient was observed? This seems like a possibility given that Dpp localized to both compartments. Is it feasible to knock down both pathways simultaneously to see how this impacts Dpp?

9) Figure 9 and Discussion.

i) Consider including additional diagrams in Fig 9 to visualize what is happening in some of the mutant backgrounds with respect to signaling and Dpp localization.

ii) (Repeated from above) Is the effect of ESCRT rnai on pMad greater than rab5 rnai? If so, why would that be with respect to the provided model.

As for all my reviews this is signed by David Fay

**Have all data underlying the figures and results presented in the manuscript been provided?**

Reviewer #1: **No: ** I did not see a spreadsheet with the data in this first version of the manuscript.

Reviewer #2: **No: ** I didn't see any info regarding raw data. Not sure what is required though.

PLOS authors have the option to publish the peer review history of their article (what does this mean? ). If published, this will include your full peer review and any attached files.

**Do you want your identity to be public for this peer review?** For information about this choice, including consent withdrawal, please see our Privacy Policy .

Reviewer #1: No

Reviewer #2: **Yes: ** David Fay

**Figure resubmission:****Reproducibility:** To enhance the reproducibility of your results, we recommend that authors of applicable studies deposit laboratory protocols in protocols.io, where a protocol can be assigned its own identifier (DOI) such that it can be cited independently in the future. Additionally, PLOS ONE offers an option to publish peer-reviewed clinical study protocols. Read more information on sharing protocols at https://plos.org/protocols?utm_medium=editorial-email&utm_source=authorletters&utm_campaign=protocols

---

## [Decision Letter · Decision Letter 1]

PGENETICS-D-24-00998R1

Shaping and interpretation of Dpp morphogen gradient by endocytic trafficking

PLOS Genetics

Dear Dr. Shinya Matsuda

Thank you for submitting your manuscript to PLOS Genetics. After careful consideration, we feel that it has merit but does not fully meet PLOS Genetics's publication criteria as it currently stands. Therefore, we invite you to submit a revised version of the manuscript that addresses the points raised during the review process.

Please submit your revised manuscript within 30 days. If you will need more time than this to complete your revisions, please reply to this message or contact the journal office at plosgenetics@plos.org. Please include the following items when submitting your revised manuscript:

We look forward to receiving your revised manuscript.

Kind regards,

Louise Cheng

Academic Editor

PLOS Genetics

Fengwei Yu

Section Editor

PLOS Genetics

Aimée Dudley

Editor-in-Chief

PLOS Genetics

Anne Goriely

Editor-in-Chief

PLOS Genetics

**Reviewers' comments:**

Reviewer's Responses to Questions

**Comments to the Authors:**

Reviewer #1: The authors have addressed my comments in their revised version of the manuscript. Thank you.

I only have two minor comments:

1. I think the Supplementary Table is helpful, but I couldn’t see a reference to it in the text – perhaps at the start of the Discussion?

2. page 9, line 26 needs rewording, ie: ‘we speculate that loss of Rab7 causes increased association of Dpp and Tkv with ILVs….’, since Dpp is likely to be outside ILVs not in them.

Reviewer #2: The authors have done a very good job of addressing prior concerns and the manuscript is substantially improved. It's a very nice story. David Fay

**Have all data underlying the figures and results presented in the manuscript been provided?**

Reviewer #1: Yes

Reviewer #2: Yes

PLOS authors have the option to publish the peer review history of their article (what does this mean? ). If published, this will include your full peer review and any attached files.

**Do you want your identity to be public for this peer review?** For information about this choice, including consent withdrawal, please see our Privacy Policy .

Reviewer #1: No

Reviewer #2: **Yes: ** David S. Fay

**Figure resubmission:**
---

## [Editor Report · Decision Letter 2]

Dear Dr Masuda,

We are pleased to inform you that your manuscript entitled "Shaping and interpretation of Dpp morphogen gradient by endocytic trafficking" has been editorially accepted for publication in PLOS Genetics. Congratulations!

Yours sincerely,

Louise Cheng

Academic Editor

PLOS Genetics

Fengwei Yu

Section Editor

PLOS Genetics

Aimée Dudley

Editor-in-Chief

PLOS Genetics

Anne Goriely

Editor-in-Chief

PLOS Genetics

Comments from the reviewers (if applicable):

**Data Deposition**

http://datadryad.org/submit?journalID=pgenetics&manu=PGENETICS-D-24-00998R2

**Press Queries**

---

## [Editor Report · Acceptance letter]

PGENETICS-D-24-00998R2

Shaping and interpretation of Dpp morphogen gradient by endocytic trafficking

Dear Dr Matsuda,

We are pleased to inform you that your manuscript entitled "Shaping and interpretation of Dpp morphogen gradient by endocytic trafficking" has been formally accepted for publication in PLOS Genetics! Your manuscript is now with our production department and you will be notified of the publication date in due course.

With kind regards,

Olena Szabo

PLOS Genetics

On behalf of:
